# Attention is Not Always Needed: Attention Sink Forges a Native MoE in Attention Layers

## Abstract

Large Language Models (LLMs) often assign disproportionate attention to the first token, a phenomenon known as the *attention sink*. Several recent approaches aim to address this issue, including *Sink Attention* in GPT-OSS and *Gated Attention* in Qwen3-Next. However, a comprehensive analysis of attention sink is lacking, and the necessity of eliminating it remains unclear. In this work, we provide both theoretical and empirical evidence showing that attention sink emerges as a mechanism to resolve forced attention, yet it limits the model's expressive capacity. By analyzing the connection between attention sink and *Gated Attention*, we demonstrate that attention sink implicitly constructs a native Mixture-of-Experts (MoE) within attention layers. This insight reveals why only a fixed subset of attention heads contributes to generation, which closely resembles the expert collapse problem encountered in MoE. To enhance the utilization balance of attention heads, we propose a sink-aware training algorithm with an auxiliary load balancing loss designed for attention layers. We hope this study offers a new practical perspective on *attention sink* and *Gated Attention*, and encourages further exploration of how to leverage the inherent MoE mechanisms within attention layers.

## 1 Introduction

StreamingLLM (Xiao et al., 2023) revealed that large language models (LLMs) often assign high attention weight to the first token, regardless of its semantic relevance. This phenomenon, termed attention sink, has been widely applied, including KV cache optimization (Ge et al., 2023; Wu & Tu, 2024; Su & Yuan, 2025), long-context generation (Xiao et al., 2024; Yang et al., 2025b; Fu et al., 2025), pruning(Sandoval-Segura et al., 2025; Shin et al., 2025), quantization (Liu et al., 2024; Huang et al., 2024), and click-through rate prediction (Li et al., 2025) (see Appendix A for details).

While many works have studied attention sink (Guo et al., 2024a; Barbero et al., 2025; Qiu et al., 2025; Agarwal et al., 2025) (see Appendix A for details), a comprehensive analysis of attention sink is still lacking. Two central questions remain to be addressed: Why does attention sink emerge? Why and how should attention sink be eliminated? For the first question, Guo et al. (2024a) explains it through the active-dormant mechanism of attention heads, while Barbero et al. (2025) attributes it to the need to avoid over-mixing. Our analysis approaches the issue from a more fundamental angle, showing that it arises from the normalization property of softmax (see Section 2.1 for details). For the second question, a variety of methods have been proposed to eliminate attention sink (Zuhri et al., 2025; Qiu et al., 2025; Agarwal et al., 2025). However, these works do not explain why such elimination is necessary. Our analysis shows that attention sink reduces the precision of query-to-key selection and thereby restricts the model's expressive capacity (see Section 2.2 for details).

Existing methods for eliminating attention sinks can be divided into two categories, as illustrated in Figure 1. The first modifies the softmax operation, exemplified by *Sink Attention* in GPT-OSS (Agarwal et al., 2025). The second introduces a gating mechanism, represented by *Gated Attention* in Qwen3-Next (Qiu et al., 2025). These two distinct approaches motivate us to explore their underlying connection with attention sinks. Surprisingly, our analysis reveals that attention weight on the sink in *Vanilla Attention* and *Sink Attention* functions as an implicit gating factor, which corresponds directly to the explicit gating factor in *Gated Attention*. This finding suggests that attention sink forges a native mixture-of-experts (MoE) within attention layers (see Section 3 for details).

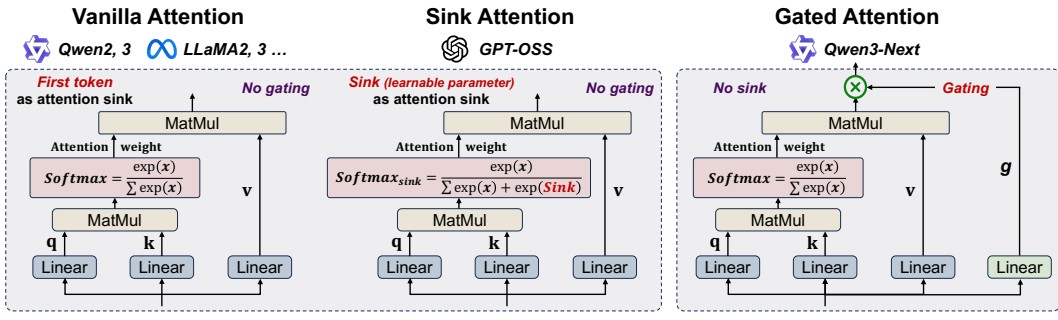

Figure 1: **Vanilla Attention** used in most open-source models. **Sink Attention** used in GPT-OSS with a learnable bias *sink*, which is added to the Softmax denominator. **Gated Attention** used in Qwen3-Next with a gating factor, which is applied to the output of each attention head.

Furthermore, we observe that in traditional models only a fixed subset of heads consistently contributes to generation, while the others remain inactive. Similar observations have also been reported in previous studies (Xiao et al., 2024; Sandoval-Segura et al., 2025). Based on the analysis that attention layers already exhibit a native MoE structure, this phenomenon aligns with the well-known issue of expert collapse in MoE models, where the model repeatedly relies on a small number of dominant experts. To address this problem, we introduce an auxiliary load balancing loss that leverages the attention sink and is specifically designed for attention layers. The proposed method encourages more balanced utilization of all heads and ultimately yields notable improvements in model performance (see Section 4 for details). Our main contributions are summarized as follows:

- We empirically and theoretically analyze the origin of attention sink and argue that assigning first token as the attention sink restricts the model's expressive capacity.
- We reveal that attention weight on attention sink works as an implicit gating factor, demonstrating the equivalence among *Vanilla Attention*, *Sink Attention*, and *Gated Attention*. Therefore, attention sink forges a native MoE within attention layers.
- To enhance the utilization balance of attention heads, we propose a sink-aware training algorithm with an auxiliary load balancing loss designed for attention layers.

## 2 AVOIDING FORCED ATTENTION OF SOFTMAX BY SINK TOKEN

### 2.1 WHY DOES ATTENTION SINK EMERGE?

In Transformer attention layers, the softmax function plays a critic role. Originating from statistical mechanics, the softmax function was introduced by Ludwig Boltzmann to describe the probability distribution of microstates in a system: $p_i = \frac{e^{-e_i/kT}}{\sum_{j=1}^{M} e^{-e_j/kT}}$. The denominator, known as the partition function, normalizes the distribution so that the probabilities across all states sum to 1. This property is fundamental to any probability distribution.

In the context of attention, a natural question arises: is such normalization always necessary? In other words, must every attention head distribute its attention weights across all tokens? The answer is no. Forced attention is not an absolute necessity. Therefore, **under softmax, each head must find a way to bypass this enforced allocation of attention**, enabling more flexible representations.

For each attention head, the output is computed as a weighted sum of the *value* vectors. It is essential that the sum of attention weights equals 1.

$$A_{t,j}^{l,h} = \text{Softmax}\left(\frac{q_t^{l,h} k_j^{l,h\top}}{\sqrt{d_h}}\right), \ t \geq j, \qquad \sum_{j=0}^{t} A_{t,j}^{l,h} = 1, \qquad O_t^{l,h} = \sum_{j=0}^{t} A_{t,j}^{l,h} \cdot v_j^{l,h}. \quad (1)$$

where $q_t^{l,h}, k_t^{l,h}, v_t^{l,h}$ are the *query*, *key*, and *value* corresponding to the $t$-th token in the $h$-th head of the $l$-th layer, and $A_{t,j}^{l,h}$ is the attention weight of the $t$-th token with respect to the $j$-th token in the $h$-th head of the $l$-th layer. The same applies in subsequent discussions.

Table 1: Performance of Qwen3-8B and Qwen3-14B under Different Settings: none (no zeroing of the first-token *value*), all (zeroing the first-token *value* across all heads), and $\tau$=0.75 (zeroing the first-token *value* for heads with $\alpha_{sink}^{l,h}$ above 0.75, as defined in Equation 5).

| Model | Method | ARC | Hella Swag | MMLU | GSM8K | BBH | BoolQ | ToxiGen | TQA | Human Eval | AVG |
|---|---|---|---|---|---|---|---|---|---|---|---|
| Qwen3 8B | None | 68.96 | 74.97 | 75.42 | 87.64 | **79.31** | 86.73 | 45.96 | 50.24 | **64.02** | 70.36 |
| | All | **69.82** | 61.02 | 50.27 | 87.57 | 76.17 | 76.48 | 42.45 | 50.33 | 62.82 | 64.10 |
| | $\tau$=0.75 | 67.53 | **76.32** | **75.46** | **87.87** | 79.24 | **86.89** | **46.06** | **51.14** | 62.90 | **70.38** |
| Qwen3 14B | None | 71.49 | 78.84 | **79.05** | **82.41** | 41.88 | 89.30 | 47.13 | 56.06 | 56.1 | 66.91 |
| | All | **73.52** | 75.76 | 74.67 | 81.73 | **46.74** | 87.28 | 45.00 | 53.51 | 54.88 | 65.90 |
| | $\tau$=0.75 | 71.66 | **79.78** | **79.05** | 81.80 | 41.88 | **89.35** | **47.23** | **56.37** | **56.71** | **67.09** |

Consider the scenario in which an attention head requires only partial focus, or no focus at all. In this case, part of the normalized attention weights becomes redundant. This redundancy can be eliminated only if the *value* vectors corresponding to the unnecessary attention weights are set to zero, thereby ensuring that they make no contribution to the final output.

However, producing an exact zero vector for the *value* is difficult in practice. During generation, activations from the previous layer vary significantly, and after projection through the value matrix $W_V$, it is unlikely that a token's *value* becomes strictly zero. A practical compromise is to use a token whose *value* vector satisfies two relaxed conditions: **(1) Its *value* vector approaches zero, carrying little semantic information**, so its contribution is negligible even if included in the output. **(2) Its position allows all tokens to attend to it during generation.** For example, if such a token appeared only at the end, earlier tokens would be unable to use it.

The first token naturally fulfills these requirements. Due to the attention mask, it carries little semantic content, and it remains attendable by all tokens. As shown in Figure 2, the *value* vector of the first token approaches zero compared to other tokens, enabling it to absorb redundant attention without influencing the output. Experiments are conducted on 500 GSM8K samples, and more results across models and layers are provided in Appendix B.

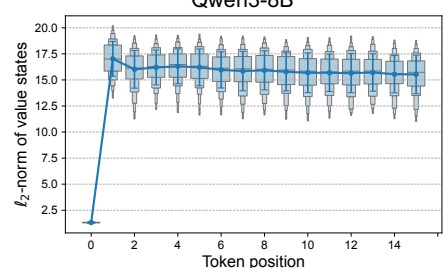

Figure 2: $\ell_2$-norm of token *values*.

To confirm that the *value* of first token does not store useful information, we performed the following experiment. After generating the KV cache for the first token in the prompt, we set its *value* vector to zero under two conditions: (1) across all heads, and (2) only for the heads showing the strongest sink effect, i.e., those with the highest first-token attention weights (see Equation 5). The results in Table 1 show that, even without additional training, zeroing out the first token's *value* vector yields performance comparable to the baseline, and in some cases even leads to improved performance.

This analysis suggests that **the first token acts as an attention sink, mitigating the forced attention of softmax by absorbing redundant attention mass.**

## 2.2 WHY AND HOW TO ELIMINATE ATTENTION SINK?

Prior works (Zuhri et al., 2025; Miller, 2023; Agarwal et al., 2025; Qiu et al., 2025; Darcet et al., 2023) have shown that assigning first token as the attention sink produces two adverse effects: (1) It creates pronounced activation outliers, which hinder quantization (Liu et al., 2024; Su & Yuan, 2025); (2) It restricts model expressiveness, making it harder to pay attention to the right tokens Zuhri et al. (2025); Qiu et al. (2025). The first issue arises because some attention heads carry redundant mass (Sandoval-Segura et al., 2025). These heads assign a large attention weight to the first token, which yields outliers in the attention weights.

For the second issue, we find that it arises from the geometric constraints imposed by the sink. As shown in Figure 3(a), the high attention weight for the first token is achieved through a high cosine similarity $\cos(\mathbf{q}_t^{l,h}, \mathbf{k}_0^{l,h})$, so that subsequent tokens must keep their *query* vectors close to the *key* of the first token. In practice, to maintain a large value of $\mathbf{q}_t^{l,h}(\mathbf{k}_0^{l,h})^\top$, the *query* vectors of

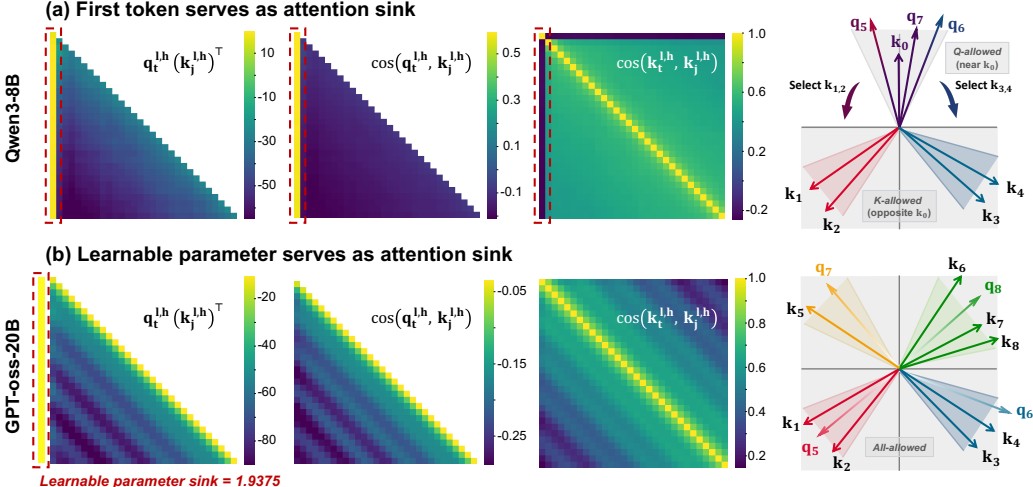

Figure 3: $\mathbf{q}_t^{l,h}(\mathbf{k}_j^{l,h})^\top$, $\cos(\mathbf{q}_t^{l,h}, \mathbf{k}_j^{l,h})$ and $\cos(\mathbf{k}_t^{l,h}, \mathbf{k}_j^{l,h})$ during generation across models. (a) Qwen3-8B (*Vanilla Attention*): $\mathbf{q}_t$ stays near $\mathbf{k}_0$ while $\mathbf{k}_t$ for $t \neq 0$ lie opposite to $\mathbf{k}_0$, making the selection of relevant keys by queries difficult. (b) GPT-OSS-20B (*Sink Attention*): directions of $\mathbf{q}_t$ and $\mathbf{k}_t$ are more flexible, enabling more precise query–key matching.

later tokens align with $\mathbf{k}_0^{l,h}$, preserving a small angle between them. Meanwhile, the *key* vectors of other tokens tend to point in the opposite direction to distinguish themselves from the first token. Consequently, during generation, all $\mathbf{q}_t$ are effectively *pinned* near $\mathbf{k}_0$, while all $\mathbf{k}_t$ for $t \neq 0$ are repelled from $\mathbf{k}_0$. As shown in Figure 3(a), they are allowed to appear only within the gray region. For a *query*, when selecting the *key* of a token (i.e., assigning a larger attention weight to that token), the choice is restricted to angular shifts within the gray region, rather than being aligned exactly with the corresponding *key* vector direction. **This geometry makes it harder for a *query* to select among competing *keys*, reducing the precision of attention allocation.** The model's expressive capacity is therefore limited, as the mechanism struggles to focus on the most relevant tokens.

Fortunately, many studies have sought to eliminate the attention sink. These methods can be grouped into two classes: (1) approaches that **modify the softmax activation to remove normalization** (Miller, 2023; Zuhri et al., 2025; Agarwal et al., 2025); (2) approaches that **introduce gating mechanisms within the attention module** to enable flexible allocation of attention (Qiu et al., 2025). Both classes are effective for a common reason: they relax the forced attention induced by normalization in other way.

The first class includes Softpick (Zuhri et al., 2025), which replaces the standard softmax with a non-normalizing variant. Another example is the modification in Miller (2023), where a constant 1 is added to the denominator. A closely related variant appears in *Sink Attention* of GPT-OSS (Agarwal et al., 2025), which augments the denominator with a learnable parameter *sink*:

$$\mathrm{Softmax}_1(x)_i = \frac{\exp(x_i)}{\sum_j \exp(x_j) + \mathbf{1}}, \qquad \mathrm{Softmax}_{\mathrm{sink}}(x)_i = \frac{\exp(x_i)}{\sum_j \exp(x_j) + \exp(\mathbf{sink})}. \quad (2)$$

By relaxing the hard normalization, $\mathrm{Softmax}_{\mathrm{sink}}$ removes the requirement that attention weights sum to one and therefore eliminates the tendency for the first token to act as a sink.

It should be noted that GPT-OSS still contains attention sink, but it is no longer tied to the first token. The learnable parameter *sink* functions as a synthetic sink and is effectively analogous to $\mathbf{q}_t^{l,h}(\mathbf{k}_0^{l,h})^\top$ in the vanilla formulation. It can be viewed as having an associated *value* of zero, which aligns with the analysis in Section 2.1 that the sink token should contribute a zero *value*.

Using a learnable *sink* as the attention sink in GPT-OSS offers several advantages over using the first token. As shown in Figure 3(b), the influence of $\mathbf{k}_0$ is avoided. **The *query* and *key* vectors of all tokens become more flexible.** As a result, attention can be allocated with greater precision, and the model exhibits stronger expressivity.

The second class of methods introduces gating mechanisms into the attention layer, as in Gated Attention (Qiu et al., 2025). As shown in Figure 1, a head-specific scalar gate is applied to the

output of each head to control how much that head contributes to the multi-head output, formalized in Equation 3. By modulating each head through this gate, the design relaxes forced attention and eliminates the attention sink phenomenon.

The two adverse effects caused by using the first token as an attention sink are also solved by gated attention. However, this benefit comes with extra cost: a routing module and additional parameters are required to produce the gating factors. Section 3.1 will show a formal link between gating and attention sink mechanism, providing a unified perspective that connects these approaches.

> **Key Insight:** *First token acts as an attention sink, mitigating the forced-attention of softmax. Using first token as the attention sink reduces the precision of query-to-key selection.*

## 3 ATTENTION SINK FORGES A NATIVE MoE IN ATTENTION LAYERS

### 3.1 ATTENTION SINK SERVES AS A GATING FACTOR

*Gated Attention* (Qiu et al., 2025) introduced approaches that integrate gating mechanisms into attention layers, thereby embedding the concept of MoE. These approaches, however, rely on an additional routing network. This naturally raises a central question: **is the MoE concept already inherent in the attention layer itself?**

The MoE framework consists of two key components. The first is a set of independent modules acting as experts. The second is a routing mechanism that selects among these modules (Shazeer et al., 2017; Fedus et al., 2021). The attention layer naturally satisfies the first condition, since its multiple heads already function as independent experts. Regarding the second condition, we find that the attention sink provides an equivalent gating factor.

To establish this equivalence, we first clarity the relation between *Vanilla Attention* and the *Sink Attention* mechanism in GPT-OSS. As discussed in Section 2.1, in *Vanilla Attention* the first token acts as the attention sink, with its corresponding $\mathbf{v}_0^{l,h}$ approaching zero. **Thus, $A_{t,0}^{l,h}$ in *Vanilla Attention* is equivalent to $\frac{\exp(sink)}{\sum_j \exp(x_j) + \exp(sink)}$ in *Sink Attention*, unified as $A_{t,sink}^{l,h}$.** In the following discussion with *Gated Attention*, we use the *Sink Attention* form for clearer token indexing, though the analysis applies equally to *Vanilla Attention*. In *Sink Attention*, the attention weights satisfy : $A_{t,sink}^{l,h} + \sum_{j=0}^{t} A_{t,j}^{l,h} = 1$ (with indices shifted by +1 in *Vanilla Attention*).

In *Gated Attention*, the attention weights satisfy: $\sum_{j=0}^{t} \tilde{A}_{t,j}^{l,h} = 1$ (where we use $\tilde{A}$ to represent the attention weights in Gated Attention), and the study shows that inserting a gating factor after each head yields the best performance(Qiu et al., 2025), formalized as follows:

$$G^{l,k} = \sigma\left(x^l W_\theta^{l,k}\right), \quad W_\theta^{l,k} \in \mathbb{R}^{d_{\text{model}} \times 1}, \quad O^l = \text{Concat}(G^{l,1} O^{l,1}, \cdots, G^{l,h} O^{l,h}) W_O^l \quad (3)$$

where $O_k$ is the output of head $k$, $G_k$ is its gating factor, and $\sigma(\cdot)$ denotes the sigmoid function.

Assuming $\mathbf{q}_t^{l,h}(\mathbf{k}_j^{l,h})^\top$ remains unchanged across architectures, the relative proportions of weights after softmax should also remain consistent: $\frac{A_{t,i}^{l,h}}{A_{t,j}^{l,h}} = \frac{\tilde{A}_{t,i}^{l,h}}{\tilde{A}_{t,j}^{l,h}}, \quad \forall i, j.$

It follows that:

$$G_{sink}^{l,h} = \frac{A_{t,i}^{l,h}}{\tilde{A}_{t,i}^{l,h}} = \frac{\sum_{j=0}^{t} A_{t,j}^{l,h}}{\sum_{j=0}^{t} \tilde{A}_{t,j}^{l,h}} = \frac{1 - A_{t,sink}^{l,h}}{1}.$$

$$O_{gated}^{l,h} = G_{gated}^{l,h} \cdot \sum_{j=0}^{t} \tilde{A}_{t,j}^{l,h} v_j^{l,h} \quad \Longleftrightarrow \quad O_{sink}^{l,h} = \sum_{j=0}^{t} A_{t,j}^{l,h} v_j^{l,h} = G_{sink}^{l,h} \cdot \sum_{j=0}^{t} \tilde{A}_{t,j}^{l,h} v_j^{l,h}. \quad (4)$$

In ungated vanilla attention, the sink weight naturally serves as a gating factor. This is referred to as the **implicit gating factor**. After transformation, the explicit form $G_{sink}^{l,h} = 1 - A_{t,sink}^{l,h}$ is termed the **explicit gating factor**. This equivalence shows that the gating factor, a central element of MoE, already exists in the attention layer. Therefore, **attention sink forges a native MoE within attention layers**.

Figure 4: The head-level metric $\alpha_{sink}^{l,h}$ across different models, defined in Equation 5. A larger $\alpha_{sink}^{l,h}$ indicates lower activation of the corresponding head.

## 3.2 ACTIVE HEADS AS UNIFIED CARRIERS OF LOCAL AND GLOBAL CONTEXT

The transition from implicit to explicit gating highlights the token-level gating mechanism. For each head during generation, the activation level can be defined as:

$$\alpha_{sink}^{l,h} = \frac{1}{T} \sum_{i=0}^{T-1} A_{i,sink}^{l,h}. \tag{5}$$

A larger $\alpha_{sink}^{l,h}$ corresponds to lower activation of that head. Figure 4 illustrates how activation varies across heads for models of different sizes and types, revealing distinct utilization patterns. For large-parameter models such as LLaMA3.3-70B and Qwen2.5-72B, the yellow region occupies a larger proportion, indicating lower head utilization. Notably, GPT-OSS-20B shows relatively less yellow, suggesting higher overall head utilization.

Following prior studies (Sandoval-Segura et al., 2025; Guo et al., 2024a) and Figure 4, heads with high attention weight on the sink are referred to as dormant heads. Their outputs are near zero and their contribution to generation is negligible. The analysis therefore focuses on the remaining active heads.

A key observation from experiments is that the same active heads are responsible for capturing both local information and global information. Each active head can be evaluated along two complementary dimensions that quantify its tendency toward local focus and global coverage.

The degree of local focus is measured by

$$rs := recent\ token\ size, \quad \alpha_{\text{local}}^{l,h} = \frac{1}{T-rs+1} \sum_{i=rs}^{T} \sum_{j=i-rs+1}^{i} A_{i,j}^{l,h} \tag{6}$$

A larger $\alpha_{\text{local}}^{l,h}$ indicates stronger emphasis on nearby tokens.

The degree of global coverage is measured by the dispersion of the attention distribution around the uniform baseline:

$$\mu_i^{l,h} = \frac{1}{i} \sum_{j=1}^{i} A_{i,j}^{l,h} = \frac{1}{i}, \quad i = 1, \dots, T, \quad \overline{\sigma}^{2(l,h)} = \frac{1}{T} \sum_{i=1}^{T} \left[ \frac{1}{i} \sum_{j=1}^{i} \left( A_{i,j}^{l,h} - \frac{1}{i} \right)^2 \right] \tag{7}$$

A smaller $\overline{\sigma}^{2(l,h)}$ reflects a flatter pattern and thus stronger capture of global information.

From Figure 5, it can be observed that most active heads, namely those with smaller $\alpha_{sink}^{l,h}$ in Figure 3, demonstrate strong ability to capture both local and global information. Another noteworthy trend is that heads in the first few layers show higher activation level and stronger global capacity. This pattern is consistently found across many models(see Appendix C for details). The attention maps also illustrate this behavior: heads in the first few layers tend to capture global information, with the first layer being particularly pronounced.

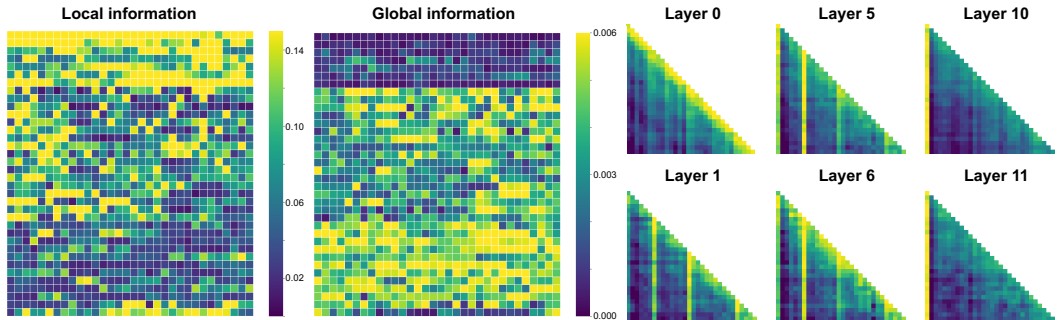

Figure 5: The head-level metrics $\alpha_{\text{local}}^{l,h}$ and $\overline{\sigma}^{2(l,h)}$ for the Qwen3-8B model, defined in Equation 6 and Equation 7, together with the $\mathbf{q}_t^l(\mathbf{k}_j^l)^\top$ attention maps across different layers. A larger $\alpha_{\text{local}}^{l,h}$ indicates stronger capture of local information, while a smaller $\overline{\sigma}^{2(l,h)}$ reflects stronger capture of global information.

An analysis based on receptive field theory provides further support for this phenomenon. The first layer in the model receives activations containing only the information of the current token. If it fails to capture global context, the second layer will again process activations dominated by the current token without access to a broader receptive field. As a result, the heads in the earlier layers play a crucial role in enabling the model to aggregate global information.

> **Key Insight:** *Attention weight on the sink functions as an implicit gating factor. The first few layers of the model plays a significant role in generation.*

## 4 SINK-AWARE TRAINING TO ADDRESS EXPERT COLLAPSE

### 4.1 EXPERT COLLAPSE WITHIN ATTENTION HEADS

From Figure 4, it can be observed that many heads in several models consistently assign high attention weights to the attention sink. The formula defined in Equation 5 applies to the entire generation, indicating that these heads remain inactive across all tokens throughout the process.

As shown in Figure 6, different types of generation tasks were tested across various models, including the GSM8k dataset for mathematics and English, and subsets of the C-Eval dataset for humanities and Chinese. These two datasets represent two contrasting domains. It can be observed that, across different types of datasets, the activation level of the same head remains essentially unchanged. The results reveal that the activated heads are largely fixed, while dormant heads do not contribute to any of the generation tasks. This pattern is consistently observed in mainstream open-source models such as Qwen3, LLaMA3, and GPT-OSS. Previous study (Xiao et al., 2024; Sandoval-Segura et al., 2025) also demonstrates that the heads playing a central role in a model are largely fixed (see Appendix A.5 for details).

Based on these observations and experiments, and in reference to existing research on MoE (Shazeer et al., 2017; Fedus et al., 2021), we can demonstrate the occurrence of **expert collapse within attention heads**. Expert collapse in MoE networks refers to the tendency of the routing mechanism to select a fixed subset of experts. In the context of attention layers, this translates into selecting a fixed subset of attention heads.

This phenomenon typically arises in the early stages of training, when a few experts are consistently activated. These experts gradually acquire stronger capabilities, causing the routing mechanism to favor them more heavily in later stages. As a result, the remaining experts are insufficiently trained, leading the model to depend disproportionately on a few dominant experts while others become effectively irrelevant.

Within attention heads, the most significant drawback of expert collapse is the waste of computational resources. Many heads consume substantial training and inference costs, however, without producing meaningful contributions. As illustrated in Figure 4, the proportion of dormant heads in some models is remarkably high. Furthermore, Figure 4 demonstrates that the issue of attention

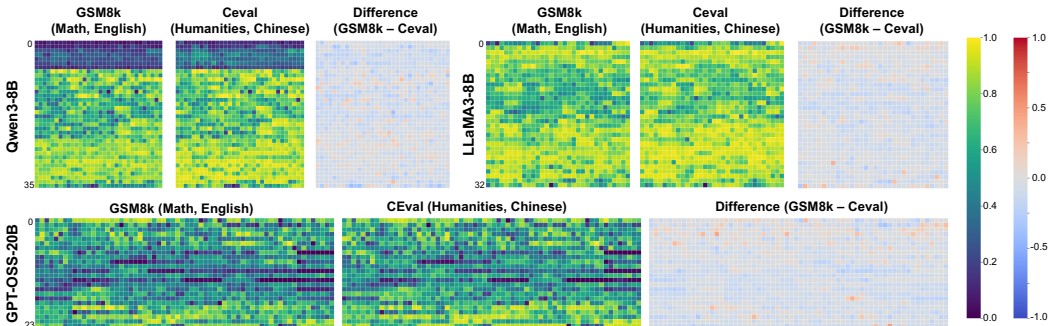

Figure 6: Comparison of $\alpha_{sink}^{l,h}$ across different models on the GSM8k and C-Eval datasets, together with their difference defined as diff = $\alpha_{sink,gsm8k}^{l,h} - \alpha_{sink,ceval}^{l,h}$. A higher $\alpha_{sink}^{l,h}$ value indicates weaker activation of the corresponding head. A diff value close to zero suggests that the head exhibits similar activation levels across the two datasets. GSM8k consists of mathematical and English problems, while C-Eval includes selected subsets from humanities and Chinese. The results show that head activation remains largely consistent across datasets of different types.

sink becomes more pronounced in larger models, where waste and idleness among attention heads grow increasingly severe. Sandoval-Segura et al. (2025) also show that larger models tend to have a higher proportion of heads whose outputs can be zeroed out. This suggests that, for larger models, the proportion of dormant heads is higher due to expert collapse.

## 4.2 AUXILIARY LOAD BALANCING LOSS FOR UTILIZATION OF ALL HEADS

To mitigate expert collapse in attention heads and to better exploit all heads, an auxiliary load balancing loss is introduced for fine-tuning with the goal of load balancing. Building on the MoE analysis of attention layers in Section 3.1 and drawing on prior MoE work (Shazeer et al., 2017; Fedus et al., 2021), we define the following new auxiliary load balancing loss tailored for the attention layer:

$$G_i^{l,h} = \frac{1 - A_{sink}^{l,h}}{1}, \quad \text{Imp}^{l,h} = \frac{1}{B} \sum_{x \in \mathcal{B}} G^{l,h}(x), \quad \mathcal{L}_{\text{aux}} = \lambda N_h N_L \, \text{CV}\Big(\{\text{Imp}^{l,h}\}\Big)^2 \quad (8)$$

where $B$ equals batch size multiplied by sequence length, $N_h$ is the number of heads in each layer, and $N_L$ is the number of layers in the model. $\text{CV}(\cdot) = \frac{\text{std}(\cdot)}{\text{mean}(\cdot)}$ denotes the coefficient of variation. $\lambda$ is a coefficient, set to 0.01 in our experiments.

Since the attention sink serves as a gating factor, it is not necessary to select top-$k$ experts within an attention layer, as is typically done in the MoE for FFN layers. This removes the restriction of heads being confined to a single layer. Therefore, the auxiliary load balancing loss is defined over all attention heads across layers. It also should be emphasized that this loss is not designed to eliminate the attention sink phenomenon. Attention sink is considered a natural property that provides a gating effect and prevents forced attention, as discussed in Sections 2.1 and 3.1. The purpose of the loss is to leverage attention sink as an implicit gating factor, enabling balanced routing among heads and ensuring that all heads are actively utilized.

In practice, however, fine-tuning with this loss reveals a tendency for all heads to reduce their reliance on attention sink (see Appendix D for details). This phenomenon, which we term the head pinning effect, arises because collapsed experts are overly dominant and critical for generation. Weakening their influence during fine-tuning causes performance degradation, making their gating factors difficult to alter. As a result, when the auxiliary load balancing loss is introduced, the remaining heads align their gating behavior with these dominant heads, which reduces the overall presence of attention sink.

Therefore, adjustments are required when applying the auxiliary load balancing loss during fine-tuning. In the loss computation, the first few layers are excluded (see Section 3.2, where heads exhibit higher activation and process global information). Additionally, collapsed heads in later

Table 2: Evaluation results of Qwen3-4B, Qwen3-8B, and LLaMA3.1-8B models fine-tuned on the AceReason-Nemotron dataset with different loss functions. Here, *Base* denotes fine-tuning with only the Base Loss, while *Base+Aux* indicates fine-tuning with the additional auxiliary load balancing loss as defined in Equation 9.

| Model | Fine-tuning Loss | GSM8k | MATH-500 | ARC | MuSR | Competition Math | Process Bench | AVG |
|---|---|---|---|---|---|---|---|---|
| Qwen3-4B | Base | **92.34** | 69.55 | 89.98 | 54.14 | 75.20 | 67.17 | 74.73 |
| | Base+Aux | 91.81 | **69.73** | **90.84** | **55.46** | **76.16** | **67.42** | **75.24** |
| Qwen3-8B | Base | 92.42 | 70.23 | 91.98 | 49.78 | 73.65 | 67.80 | 74.31 |
| | Base+Aux | **93.33** | **71.94** | **92.34** | **53.50** | **75.09** | **69.85** | **76.01** |
| LLaMA3.1-8B | Base | 78.85 | 54.21 | 80.41 | 44.87 | 57.03 | 23.63 | 56.5 |
| | Base+Aux | **81.18** | **56.21** | **80.60** | **46.17** | **58.52** | **31.14** | **58.97** |

layers are also removed:

$$\mathcal{S}_k = \{(l,h) \mid l = k+1, \ldots, N_L\}, \quad \mathcal{T}_\alpha = \arg\underset{m}{\mathrm{top}}\Big\{\mathrm{Imp}^{l,h} \mid (l,h) \in \mathcal{S}_k\Big\}, m = \lceil \alpha(N_L - k)N_h \rceil$$

$$\mathcal{L}_{\mathrm{aux}} = \lambda n \left[ \mathrm{CV}\Big(\big\{\mathrm{Imp}^{l,h} : (l,h) \in \mathcal{S}_k \setminus \mathcal{T}_\alpha\big\}\Big) \right]^2, n = \lfloor (1-\alpha)(N_L - k)N_h \rfloor \tag{9}$$

where $k$ is the number of the first few layers that exhibit high activation, e.g., $k = 7$ in Qwen3-8B. $\alpha$ is the proportion of removed collapsed heads relative to the heads in the subsequent layers, set to 0.2 in our experiments. Following the terminology of DeepSeek-MoE (Dai et al., 2024), the excluded collapsed heads can be viewed as *Shared Experts*, while the remaining heads participating in the auxiliary load balancing loss are treated as *Routed Experts*.

Using this refined loss, we fine-tune the models Qwen3-4B, Qwen3-8B (Yang et al., 2025a), and LLaMA3.1-8B (Dubey et al., 2024) for one epoch on the AceReason-Nemotron dataset(Liu et al., 2025) to evaluate reasoning performance. Fine-tuning was implemented with the LLaMA-Factory framework (Zheng et al., 2024b) using LoRA, and all experimental settings were kept identical except for the loss function. Evaluation was carried out using the EvalScope framework (Team, 2024), with benchmarks including GSM8k (Cobbe et al., 2021), MATH500 (Hendrycks et al., 2021), ARC (Clark et al., 2018), MuSR (Sprague et al., 2023), Competition-Math (Hendrycks et al., 2021), and Process-Bench (Zheng et al., 2024a). All experiments were run on NVIDIA A100 GPUs and RTX 4090 GPUs. Detailed settings for fine-tuning and evaluation are provided in Appendix E.

As shown in Table 2, fine-tuning with the auxiliary load balancing loss yields a significant improvement in accuracy compared with the baseline. It is also evident that larger models benefit more from this loss. The reason is that in larger models, a greater portion of attention heads tends to remain underutilized, and load balancing increases their effective usage. Based on these preliminary fine-tuning experiments, together with the analysis in Section 4.1, we hold the firm belief that introducing the auxiliary load balancing loss into both pre-training and post-training stages can also bring substantial gains. Moreover, the larger the model, the more attention heads remain underutilized, and thus the greater the potential improvement from balancing their usage. In addition, many insights from prior MoE studies can be adapted to refine the auxiliary load balancing loss. All these directions call for further exploration in future work.

## 5 CONCLUSION

In this work, we conducted a detailed analysis of the origins of the attention sink phenomenon and its adverse effects. We further examined two mechanisms, *Sink Attention* and *Gated Attention*, and clarified that attention sink functions as an implicit gating factor, thereby forming a native MoE within attention layers. Moreover, motivated by the observation of expert collapse, we introduced an auxiliary load balancing loss tailored for attention layers, which encourages a more balanced utilization of attention heads. We firmly believe that this loss can also enhance model performance in other training scenarios. Finally, incorporating MoE into attention layers opens promising directions for future research. Existing studies on MoE in linear layers may provide valuable insights and methods that can be adapted to improve both the performance and efficiency of attention layers.

## ETHICS STATEMENT

This work does not involve human subjects, personally identifiable information, or sensitive data. All datasets used are publicly available and widely adopted in prior research. We have taken care to ensure that the methods and results reported in this paper do not pose foreseeable risks of harm, misuse, or unfair bias.

## REPRODUCIBILITY STATEMENT

We have made extensive efforts to ensure the reproducibility of our results. All figures and experiments presented in the paper are supported by corresponding code, which is provided in the anonymous repository at https://anonymous.4open.science/r/Attention-Sink-MoE-B514. The repository is organized to mirror the structure of the paper: code for each experimental section is placed in separate folders, corresponding directly to the experiments described in the main text. This organization allows readers to easily reproduce results for each part of the paper without ambiguity. Detailed instructions for running the experiments and reproducing the figures are included in the repository's documentation.

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

## LLM Usage Statement

During the preparation of this manuscript, a LLM (GPT-5) was employed solely for language refinement. After completing the full draft of the paper, we used the LLM to polish grammar, improve sentence clarity, and enhance overall readability. The intellectual content, research design, analysis, and all original writing were produced entirely by the authors; the LLM was not involved in generating ideas, structuring arguments, or contributing substantive content.

## A   Related Work

### A.1   Attention Sink

StreamingLLM (Xiao et al., 2023) observes that during LLM inference, the attention weights of all tokens toward the first token are significantly high, a phenomenon termed *attention sink*. This observation is particularly intriguing because, due to the presence of attention masks, the first token can only contain information about itself. Typically, this token carries limited semantic content, making its disproportionately high attention weight puzzling.

In Darcet et al. (2023), a class of tokens in images is defined as *artifacts*: tokens that appear primarily in low-informative background areas yet receive high attention weights. This phenomenon is analogous to the attention sink. However, unlike LLMs, image tokens benefit from a bidirectional attention mechanism without masking, enabling a global receptive field. Consequently, it cannot be conclusively stated that the key-value (KV) cache of these tokens contains no additional information. Indeed, Darcet et al. (2023) demonstrate that such tokens encode global information and can be used for whole-image classification tasks. Therefore, in this work, we restrict our analysis to LLMs with attention masks and exclude models based on bidirectional attention.

Yu et al. (2024) extend the definition of attention sinks to include other tokens, such as punctuation marks, which receive disproportionately high attention. However, because these tokens attend to the entire preceding sequence, it cannot be concluded that they lack semantic information. Furthermore, Chen et al. (2024) argue that punctuation tokens may serve as compressed representations of prior context rather than being semantically empty. This implies that these tokens also carry meaningful information. Hence, in this paper, only the first token is classified as an attention sink.

Gu et al. (2024) provide a detailed discussion on the conditions under which attention sinks emerge. They introduce $\mathrm{Sink}_1^\epsilon = \frac{1}{L}\sum_{l=1}^{L} \frac{1}{H}\sum_{h=1}^{H} \mathbb{I}\left(\alpha_1^{l,h} > \epsilon\right)$ to quantify attention sinks. However, their analysis does not consider head-level granularity in LLMs. In this paper, we conduct a more comprehensive head-level analysis of attention sinks.

### A.2   Leveraging Attention Sink

Many studies have leveraged attention sinks to enhance model performance (Yu et al., 2024; Sandoval-Segura et al., 2025; Han et al., 2025; Shin et al., 2025; Li et al., 2025; Su & Yuan, 2025). Yu et al. (2024) focus on the excessive attention given to the initial token and propose Attention Calibration Technique (ACT), a training-free approach that optimizes attention distributions during inference, yielding up to 7.30% accuracy improvement. Guo et al. (2024b) study the interplay between attention scores and value vector norms for token pruning, introducing Value-Aware Token Pruning (VATP), which combines these factors and achieves superior performance across several tasks.

Sandoval-Segura et al. (2025) define dormant attention heads dominated by attention sinks and demonstrate that pruning these heads has minimal impact on accuracy. Their findings reveal that such heads emerge early in pretraining and depend on input characteristics. Han et al. (2025) identify the semantically empty initial token as an attention sink and propose ZeroTuning, a training-free method that adjusts its attention distribution, improving classification accuracy by up to 11.71%. Shin et al. (2025) analyze the relationship between attention sinks and token similarity in hidden states and introduce OrthoRank, a token selection strategy that prioritizes tokens orthogonal to the sink token, outperforming layer pruning in efficiency and accuracy.

## A.3 ORIGIN OF ATTENTION SINK

Several works have sought to explain the origin of attention sinks in LLMs (Guo et al., 2024a; Zhang et al., 2025; Barbero et al., 2025).

Guo et al. (2024a) explore a broader class of extreme-token phenomena, including attention sinks, value-state drains, and residual-state peaks. Through controlled experiments on the Bigram-Backcopy task, they propose an active-dormant mechanism, where certain heads act as sinks selectively depending on the input domain. However, this activation pattern is observed only in a small subset of heads and models. Subsequent work (Xiao et al., 2024; Yang et al., 2025b; Fu et al., 2025; Sandoval-Segura et al., 2025) and our experiments suggest that the dominant heads remain fixed, aligning with the expert collapse phenomenon described in Section 4.

Zhang et al. (2025) empirically investigate how attention sinks interact with outlier features. They reveal a catch–tag–release process, in which sinks capture token sequences, perturb them with shared signals, and reintroduce them into the residual stream for later retrieval. This mechanism is shown to be necessary for tasks such as averaging, explaining its presence in LLMs. They further demonstrate that attention sinks admit low-rank parameterizations, offering insights into model compression and the effectiveness of low-rank adaptation methods.

Barbero et al. (2025) address the fundamental question of why attention sinks arise. They argue that sink tokens prevent excessive feature mixing, thereby supporting stable information propagation. Both theoretical and empirical evidence show that architectural factors such as context length, model depth, and data packing influence sink behavior. These results position attention sinks as functional components rather than artifacts, highlighting their contribution to effective learning dynamics in LLMs.

## A.4 GATED ATTENTION

In several previous works, attempts have been made to incorporate gating mechanisms within attention layers (Zhang et al., 2022; Csordás et al., 2023; Jin et al., 2024; Qiu et al., 2025; Yang et al., 2025c), following the paradigm of Mixture-of-Experts (MoE). MoE typically involves two core components: a gating mechanism and a set of experts, where each expert is responsible for distinct computations.

In *Gated Attention* (Qiu et al., 2025), the role of gating within attention was systematically studied through extensive experimentation. The key finding is that a simple yet effective modification, namely introducing a head-specific sigmoid gate after the Scaled Dot-Product Attention (SDPA), consistently improves performance. This adjustment enhances training stability, permits the use of larger learning rates, and provides better scaling behavior.

Beyond these improvements, the study devotes particular attention to the long-standing issue of the *attention sink*. Introducing query-dependent sparse gating after SDPA fundamentally alters this pattern. The gating mechanism injects input-dependent sparsity, which suppresses redundant attention allocation and effectively eliminates attention sinks.

The authors further connect this effect to training stability and long-context generalization. Sparse gating not only reduces massive activations in the hidden states but also mitigates numerical instabilities during BF16 training, thereby supporting higher learning rates. These results suggest that attention sinks are not merely an artifact of softmax normalization but a major bottleneck in scaling transformers to long contexts. By resolving this phenomenon, gated attention establishes a new perspective on the role of sparsity and opens pathways for more stable and length-generalizable architectures.

## A.5 FIXED HEADS CONTRIBUTE TO GENERATION

Previous studies provide additional evidence for this expert collapse phenomenon.

Duoattention (Xiao et al., 2024) demonstrates that the heads playing a central role in a model are largely fixed. The method first identifies the most important heads across a small number of datasets and ensures that these key heads perform full attention during inference. The remaining heads are then assigned lightweight sparse attention methods to accelerate inference. This strategy achieves

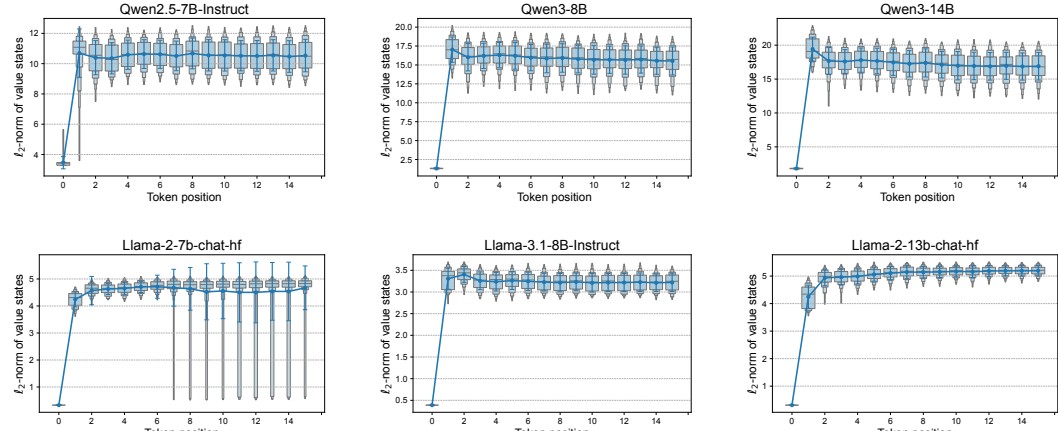

Figure 7: $\ell_2$-norm of token *values* across models.

performance comparable to full attention across all heads on different datasets. Subsequent works, such as Lserve (Yang et al., 2025b) and H2EAL (Fu et al., 2025), continue to build upon this approach, further illustrating that certain heads consistently play a significant role across all generative tasks, while some heads become less active.

In a more radical approach, Sandoval-Segura et al. (2025) show that zeroing more than 14% of a model's attention heads still preserves comparable accuracy. Their method identifies unimportant heads on a few datasets and directly zeros out their outputs in subsequent generation. These results indicate that there exists a subset of heads that are not essential for every forward pass.

## B $\ell_2$-NORM OF TOKEN VALUES

Figure 7 shows the $\ell_2$-norm of token *values* across models including Qwen2.5-7B-Instruct, Qwen3-8B, Qwen3-14B,LLaMA2-7B-chat-hf,LLaMA3-8B-Instruct,LLaMA2-13B-chat-hf, evaluated on 500 GSM8K samples. The models are divided into two categories for analysis. The first category, represented by the Qwen series, does not use ¡bos¿ as the first token. Instead, the first token is the actual input, which differs across generations. The second category, represented by the LLaMA series, starts with ¡bos¿ as the first token. The $\ell_2$-norm of token *values* at different layers is reported for each model, followed by a detailed analysis.

As shown in Figure 8, Qwen models do not use ¡bos¿ as a template. As a result, the first token can be any token, which leads to differences in the first token *value* at the first layer. This explains why the $\ell_2$-norm of the first token *value* at the first layer is not small. However, in subsequent layers, the values approach zero, leaving the overall conclusion unaffected.

As shown in Figure 9, LLaMA models adopt ¡bos¿ as the template. Unlike Qwen models, the $\ell_2$-norm of the first token *value* at the first layer also approaches zero. This indicates that the model leverages the ¡bos¿ token during training, effectively using it as an attention sink.

## C ABILITY OF LOCAL FOCUS AND GLOBAL COVERAGE

In this experiment, we evaluate $\alpha_{\text{local}}^{l,h}$ and $\overline{\sigma}^{2(l,h)}$ on 500 samples from gsm8k. The attention maps on the right are tested on one gsm8k samples, which allows observation of the attention distribution across different layers for specific tokens, and the patterns remain consistent across all samples. The results are presented in Figure 10 and Figure 11. It can be seen that active heads attend to both global and local information. The first few layers play a crucial role in acquiring global information. The attention maps further show that the first few layers focus more on global context, while subsequent layers gradually shift to particular tokens. At this stage, attention sinks begin to emerge, and in later layers the sink effect becomes severe, indicating that the heads enter a dormant state.

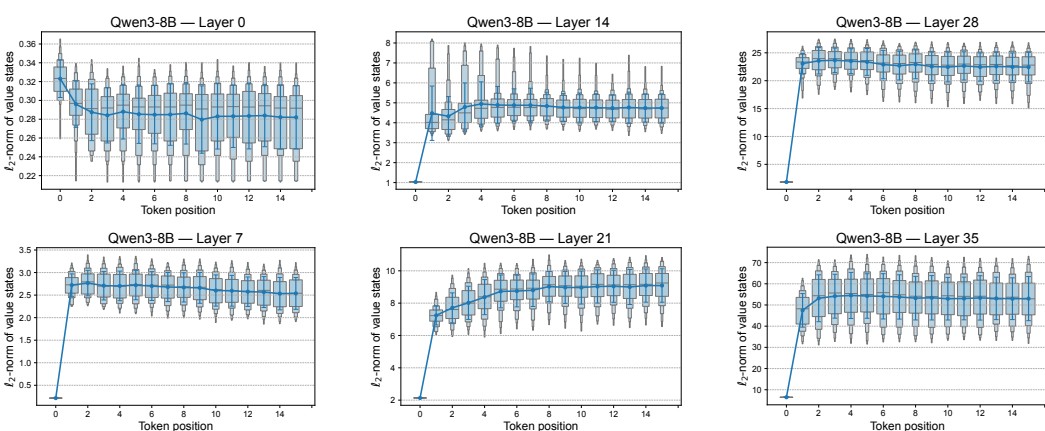

Figure 8: $\ell_2$-norm of token *values* at different layers in Qwen3-8B.

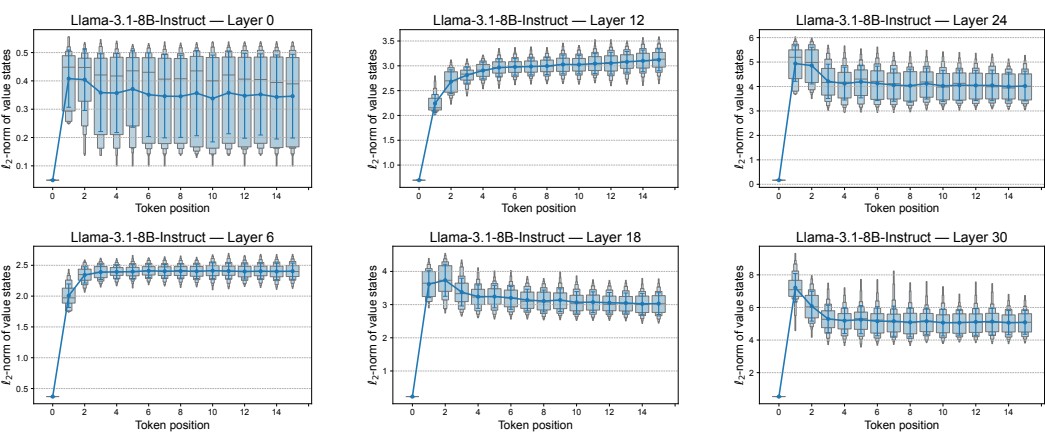

Figure 9: $\ell_2$-norm of token *values* at different layers in LLaMA3.1-8B.

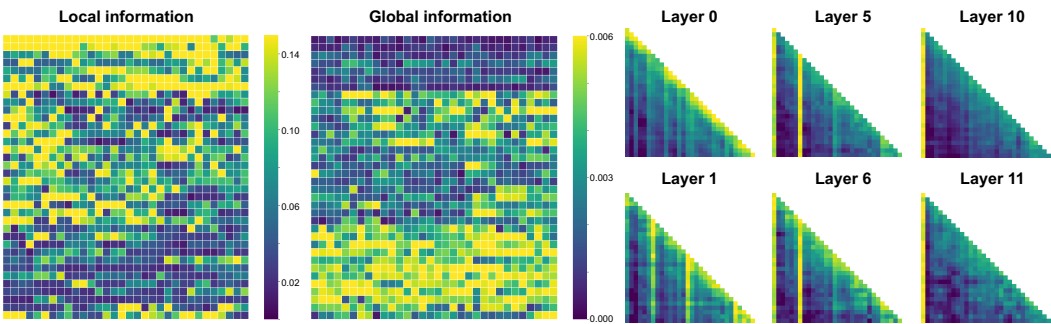

Figure 10: The head-level metrics $\alpha_{\text{local}}^{l,h}$ and $\overline{\sigma}^{2(l,h)}$ for the Qwen3-4B model, defined in Equation 6 and Equation 7, together with the attention maps across different heads. A larger $\alpha_{\text{local}}^{l,h}$ indicates stronger capture of local information, while a smaller $\overline{\sigma}^{2(l,h)}$ reflects stronger capture of global information.

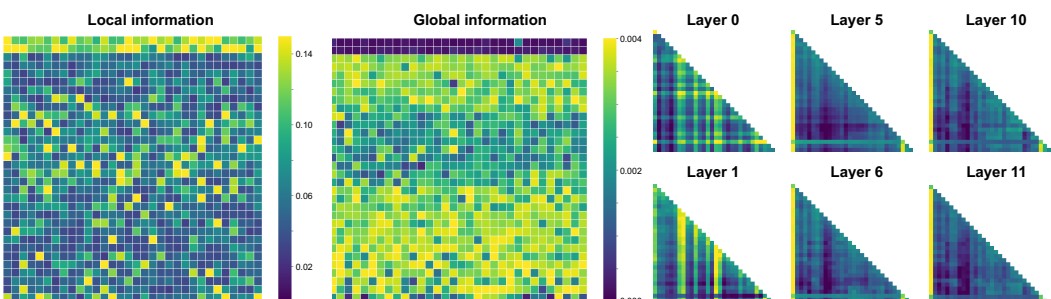

Figure 11: The head-level metrics $\alpha_{\text{local}}^{l,h}$ and $\overline{\sigma}^{2(l,h)}$ for the LLaMA2-7B-chat-hf model, defined in Equation 6 and Equation 7, together with the attention maps across different heads. A larger $\alpha_{\text{local}}^{l,h}$ indicates stronger capture of local information, while a smaller $\overline{\sigma}^{2(l,h)}$ reflects stronger capture of global information.

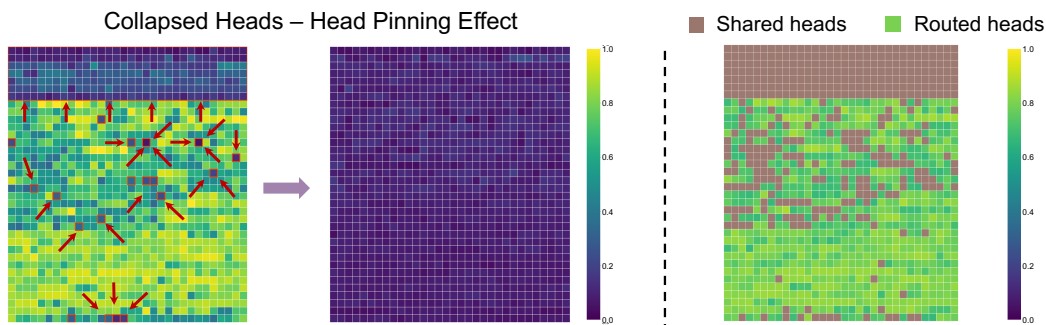

Figure 12: The left panel shows $\alpha_{sink}^{l,h}$ of Qwen3-8B before training, the middle panel shows $\alpha_{sink}^{l,h}$ after one epoch of fine-tuning with the auxiliary load balancing loss in Equation 8, and the right panel shows $\alpha_{sink}^{l,h}$ after one epoch of fine-tuning with the auxiliary load balancing loss in Equation 9.

# D    HEAD PINNING EFFECT

From Figure 12, it can be observed that after fine-tuning with the auxiliary load balancing loss in Equation 8, the attention sink across all heads largely disappears. This outcome, however, is not the intended goal. The objective is to ensure a more balanced utilization of all heads. This phenomenon, referred to as the head pinning effect, arises because collapsed experts become overly dominant and essential for generation. Reducing their influence during fine-tuning leads to performance degradation, which makes their gating factors resistant to change. Consequently, when the auxiliary load balancing loss is introduced, the remaining heads adjust their gating behavior to align with these dominant heads, thereby reducing the overall presence of attention sink. This effect is expected to occur only during fine-tuning. If the loss were applied in pre-training, similar to the MoE setting in FFN layers, such issues would likely be avoided and significant benefits could be realized.

The right panel of Figure 12 shows that fine-tuning with the auxiliary load balancing loss in Equation 9 results in a more uniform distribution of $\alpha_{sink}^{l,h}$, indicating a more balanced activation of heads within the model.

# E    EXPERIMENT SETTINGS

**Fine-tuning Setup**    Three models were fine-tuned under identical parameter configurations: Qwen3-4B, Qwen3-8B, and LLaMA-3.1-8B-Instruct. The fine-tuning was conducted using Low-Rank Adaptation (LoRA) with the LLaMA-Factory framework (Zheng et al., 2024b), with a rank of 16 and applied across all target modules. Training was executed with DeepSpeed ZeRO-3 opti-

mization, ensuring efficient distributed training. The fine-tuning utilized data from the AceReason-Nemotron dataset (Liu et al., 2025). All samples with lengths between 0 and 3072 tokens were selected. To standardize processing, the Qwen3-style and LLaMA3-style template was adopted, and a maximum cutoff length of 2048 tokens was enforced. Training was performed for one epoch with a learning rate of $1 \times 10^{-3}$. The cosine learning rate scheduler was used with a warmup ratio of 0.1. The batch size was set to 5 per device, combined with 8 gradient accumulation steps, resulting in an effective batch size of 40. Mixed-precision training with bfloat16 was enabled. For training with the auxiliary load balancing loss, the coefficient was set to $\lambda = 0.001$. All training, for each model and for each type of loss, was conducted on four NVIDIA A100 GPUs.

**Evaluation Setup**  Evaluation was conducted using the EvalScope framework (Team, 2024). The benchmarks included GSM8k (Cobbe et al., 2021), MATH500 (Hendrycks et al., 2021), ARC (Clark et al., 2018), MuSR (Sprague et al., 2023), Competition-Math (Hendrycks et al., 2021), and Process-Bench (Zheng et al., 2024a). These datasets cover a wide range of reasoning tasks, providing a comprehensive assessment of model performance. For all models, the same generation settings were applied to ensure fairness and comparability. The maximum number of generated tokens was set to 4096 to minimize the risk of truncation. Sampling was controlled with a temperature of 0.6, nucleus sampling with $p = 0.95$, and top-$k$ sampling with $k = 20$. These parameters balance output diversity and stability, and are recommended in Qwen-Report(Yang et al., 2025a). Each model was evaluated three times under identical conditions, and the highest score among the three runs was reported for each benchmark. All evaluations were performed on NVIDIA A100 and RTX 4090 GPUs.

