# OpenReview forum: "Attention is Not Always Needed: Attention Sink Forges a Native MoE in Attention Layers"
_ICLR.cc/2026/Conference — ICLR 2026 Conference Withdrawn Submission_

### Official Review · Reviewer_Xi8E · 2025-10-29

**Soundness:** 3
**Presentation:** 2
**Contribution:** 3
**Rating:** 4
**Confidence:** 5

**Summary:**

This work analyzes the “attention sink” phenomenon and argues that the sink weight functions like a head-level gate, making an attention layer in transformer block a native Mixture-of-Experts structure. Specifically, this work first elaborates that using the first token as the attention sink reduces the precision of query-to-key selection, indicating the necessity of eliminating attention sink for models’ expressive capacity enhancement. Then, this work demonstrates a formal equivalence between attention sinks and the gating factor in MoE. Moreover, this work further shows that the heads in the first few layers show higher activation level and stronger global capacity. Finally, this work reveals the occurrence of expert collapse within attention head and proving that the proportion of dormant heads is higher due to the expert collapse for larger models. To solve this expert collapse, inspired by MoE, this work finally propose an auxiliary load-balancing loss across heads with a refined variant that excludes early layers and top collapsed heads which can be regarded as the shared experts as in the MoE models. By leveraging the auxiliary load-balancing loss function, this work fine-tuning Qwen3-4B/8B and LlaMA3.1-8B model and evaluate them on six reasoning benchmarks, demonstrating modest but consistent gains overall.

**Strengths:**

1. The theoretical derivation of connecting attention sink (implicit gating factor) to an explicit gating factor is convincing, offering an interesting perspective on attention sinks and gated attention via showing the attention structure with multiple heads is inherently MoE structure. This potentially makes it possible to think attention problems as MoE routing problems.

2. In the meantime, the comprehensive analyses and empirical observations including the head-level metrics before and after training corroborate the proposed equivalence between sink mass and gating, making the claim more robust.

3. The proposed  auxiliary load-balancing loss function inspired from MoE scenario is convincing and makes sense, the fine-grained exclusion of first few years and collapsed heads is intuitive and empirically justified. The corresponding experiment results of fine-tuning existing models with the proposed loss function are convincing and align with the theoretical claims.

4. The paper is easy to read and well structured despite some statements are too strong.

**Weaknesses:**

1. The early-layer “receptive field” claim is overstated. The text asserts the first layer receives “only current-token information” and, if it fails to capture global context, the second layer will again process mostly current-token content. However, in decoder-only transformers, self-attention at layer 1 already has causal access to all prior tokens, so the statement risks conflating MLP channel information with attention context. This statement should be carefully adjusted.

2. Even though the proposed auxiliary load-balancing loss function can effectively mitigate expert collapse in attention heads, the authors report that the initial variant unintentionally reduced sink usage across the board, necessitating the refined Eq. (9), which underscores design sensitivity and potential fine-tuning instability and robustness issues for different models.

3. The evaluation seems limited. The reported gains are on AceReason-Nemotron fine-tuning and a set of reasoning benchmarks; it would strengthen claims to include long-context tasks (where sink is often exploited) and more strong reasoning tasks. Moreover, the evaluation reports the maximum of three stochastic runs is cherry-picking. It inflates scores and isn’t statistically meaningful. Report mean ± std, or use deterministic decoding (temperature=0) for final-answer benchmarks would be more convincing.

**Questions:**

1. Could you quantify how often the assumption 𝑣≈0 holds by reporting in Sec. 3.1, across layers and heads? This is important to substantiate the claimed gating equivalence.

2. To better understand the effectiveness and motivation of the loss function in Eq. (8), could you please demonstrate how sensitive are results to $\lambda$, $k$, and $\alpha$?

3. Moreover, the leveraging of CV statistic makes sense, but how does it perform compared to alternative balancing objectives like per-layer L2-to-mean?

4. Given the known link between attention sinks and quantization outliers, could you report any effects on quantization stability or activation outlier rates after fine-tuning?

---

### Official Review · Reviewer_W3GH · 2025-10-30

**Soundness:** 3
**Presentation:** 3
**Contribution:** 2
**Rating:** 4
**Confidence:** 4

**Summary:**

This paper attempts to understand attention sink phenomenon from the gating perspective. Firstly, the authors analysed the behaviours of recent open-sourced LLMs and demonstrated that first token acts as an attention sink, which is due to the forced attention of softmax. Then, through the comparisons vanilla attention, sink attention used in GPT-OSS, and Gated attention, the authors derive that attention sink serves as implicit gating factor, which forces a native MoE. Based on this understanding, the authors assume that the attention sink head experiences expert collapse, and proposed an auxiliary land balancing loss for attention scores. This new loss seems to improve the LLM performance in the continuing fine-tuning setups.

**Strengths:**

The authors provide a new perspective to understand the phenomenon of attention sink: attention could also be regarded as MoE while attention sink serves an implicit gating factor. Through this understanding, one could introduce the techniques used in the MoE community to handle attention. This part to me is novel and interesting. Through a continuing fine-tuning setup, the authors show that employing auxiliary loss similar to MoE community could bring some performance gains.

**Weaknesses:**

I think the most interesting part of this paper is to use an analogy of MoE to understand self-attention and employ MoE techniques to improve self-attention. However, the authors fail to explain this with more contents and empirical studies, especially considering some parts of this paper seem to be redundant.

From Section 2, two main messages are: (1) first token acts as attention sink, and it emerges due to normalisation in softmax, or called "forced" attention in the paper. (2) attention sink functions through the cosine similarity of queries and keys.  However,  these two messages are already covered by [1]. Additionally, [1] seems to be an early attempt to mitigate attention sink by relaxing normalisation in softmax (using sigmoid attention without normalization). The authors only cite [1] in Appendix. This not only ignores the contributions of [1], but also makes section 2 redundant. Since MoE is the most important part in this paper, I think section 2 should be a motivated part with previous literature and understanding of recent models. Of course, this reduces the contribution of this paper.

In section 3, the authors convey the message that the first few layers of the model plays a significant role in generation. This is drawn from the visualisation of Figure 5. However, I believe this message is not new to the community, like also shown in [1-3]. But I appreciate the design of local/global metrics and hope to see more analysis on receptive field theory as mentioned in this paper. Again, metric in (5) was proposed in [1] without citation.

With the analogy of MoE,  I feel that the authors fail to conduct enough empirical analysis in this part. For example, the authors only one continuing fine-tuning experiment (with LoRA) to show the performance of auxiliary loss. What are the performance of full fine-tuning? Whether the results are sensitive to k and $\alpha$ used in equation 9? The authors follow a style of mechanism understanding for storytelling. However, after fine-tuning LLMs with a new loss, they fail to conduct analysis on this new model to conclude whether the assumptions are valid.


References:\
[1] Gu et al. When attention sink emerges in language models: An empirical view. ICLR 2025.\
[2] Xiao et al. Efficient Streaming Language Models with Attention Sinks. ICLR 2024.\
[3] Chen et al. An Image is Worth 1/2 Tokens After Layer 2: Plug-and-Play Inference Acceleration for Large Vision-Language Models. ECCV 2024.

**Questions:**

Please see the weakness.

I know it may not be feasible to consider the setup of pre-training. Do you think whether the MoE analogy of self-attention could benefit pre-training?

---

### Official Review · Reviewer_z5ux · 2025-10-30

**Soundness:** 3
**Presentation:** 3
**Contribution:** 3
**Rating:** 4
**Confidence:** 3

**Summary:**

This paper provides a comprehensive analysis of the attention sink phenomenon in Large Language Models (LLMs), where the first token receives disproportionately high attention weight. The authors reframe this not just as a quirk but as a fundamental architectural feature with a direct link to Mixture-of-Experts (MoE).

**Strengths:**

The connection between attention sinks and a native MoE mechanism is an original and impactful contribution. It provides a unified theoretical lens to understand Vanilla, Sink, and Gated Attention.

The paper systematically addresses the "why" (origin), "so what" (expressiveness cost), and "what now" (MoE equivalence and solution) of attention sinks, forming a complete and logical narrative.

The findings are backed by experiments across multiple model families (Qwen, LLaMA) and scales (4B to 70B+ parameters), and evaluated on a solid set of reasoning benchmarks.

**Weaknesses:**

The auxiliary loss is only validated in a fine-tuning setting. The authors' promising claim that it would yield even greater benefits in pre-training remains a hypothesis for future work.

The performance comparison is primarily between fine-tuning with and without the proposed loss. A comparison with other state-of-the-art methods for combating head redundancy or improving attention mechanisms would further strengthen the claims.

**Questions:**

1. The authors suggest that incorporating the load-balancing loss during pre-training could be highly beneficial but might face challenges like the "head pinning effect." Based on your analysis, what are the initial conditions or optimization dynamics during pre-training that lead to expert collapse? What would be the primary challenge in applying this loss from the start of pre-training?

2. Given that attention layers constitute a native MoE and FFN layers are often explicitly converted to MoE, how should these two MoE systems be coordinated in a model that has both? Is there redundancy or synergy? Could your load-balancing loss be extended to jointly optimize both types of experts?

3. The authors argue that eliminating the sink improves attention allocation precision. Does this mean the method, particularly the load-balancing loss, could significantly enhance performance on long-context tasks?

---

### Official Review · Reviewer_HTsS · 2025-11-07

**Soundness:** 2
**Presentation:** 3
**Contribution:** 2
**Rating:** 4
**Confidence:** 4

**Summary:**

The paper analyzes attention sink as a byproduct of softmax “forced attention,” showing that the first token often has near-zero value norms and absorbs excess probability mass. It further claims an algebraic link between the sink weight and gating, arguing that attention already implements a “native MoE”. Building on head-activation diagnostics, the authors propose a sink-aware auxiliary load-balancing loss that, after one-epoch LoRA fine-tuning, improves scores on several reasoning benchmarks

**Strengths:**

- **Clear and well-presented analysis**. The paper is clearly structured, with intuitive figures and concise equations that make its argument easy to follow.

- **Comprehensive empirical coverage**. Experiments span multiple model families and reasoning benchmarks, showing that the authors tested their ideas broadly rather than on a single case.

- **Thought-provoking conceptual link**. The proposed interpretation of sink weights as implicit gating offers a unifying lens between attention mechanisms and mixture-of-experts architectures, which could inspire follow-up studies.

**Weaknesses:**

- **The claim on the cause of attention sink is not new**. The claim that attention sinks arise from softmax normalization and hurt model expressivity has already been shown in earlier studies [1, 2]. Those works provided similar empirical evidence and interpretation, so this paper largely repeats established findings rather than introducing a new explanation.

- **Key theoretical claim not well supported**. The proposed equivalence between sink attention and gated attention assumes that the relative attention ratios stay unchanged when adding gating, but this assumption is neither formally proved nor tested on a consistent setup. This weakens the central “native MoE” argument.

- **Weak statistical evidence**. Reported results are based on the best of three runs, without variance or confidence intervals. This makes it difficult to judge whether the reported improvements are consistent or just due to randomness.

- **Stability and sensitivity issues**. The proposed auxiliary loss requires manual exclusion of early layers and removal of some heads to avoid “head pinning,” suggesting that training can become unstable and sensitive to hyperparameters.

[1] Xiao, Guangxuan, et al. "Efficient streaming language models with attention sinks." arXiv preprint arXiv:2309.17453 (2023).

[2] Yu, Zhongzhi, et al. "Unveiling and harnessing hidden attention sinks: Enhancing large language models without training through attention calibration." arXiv preprint arXiv:2406.15765 (2024).

**Questions:**

What are the formal or empirical conditions under which the equivalence between sink weights and head gates holds? Could you demonstrate cases where this assumption breaks?

Could you provide mean and standard deviation over multiple runs instead of best-of-three reporting, and clarify which HumanEval metric is used?

How sensitive are the results to the hyperparameters (λ, α, k) in the auxiliary loss, and what is the computational overhead of applying it?

---

### Note · Authors · 2025-12-16

I have read and agree with the venue's withdrawal policy on behalf of myself and my co-authors.